# Comprehensive Comparative Analysis Sheds Light on the Patterns of Microsatellite Distribution across Birds Based on the Chromosome-Level Genomes

**DOI:** 10.3390/ani13040655

**Published:** 2023-02-13

**Authors:** Kaize Feng, Chuang Zhou, Lei Wang, Chunhui Zhang, Zhixiong Yang, Zhengrui Hu, Bisong Yue, Yongjie Wu

**Affiliations:** 1Key Laboratory of Bioresources and Ecoenvironment (Ministry of Education), College of Life Sciences, Sichuan University, Chengdu 610064, China; 2Sichuan Key Laboratory of Conservation Biology on Endangered Wildlife, College of Life Sciences, Sichuan University, Chengdu 610064, China

**Keywords:** birds, chromosome-level genome, microsatellite, distribution pattern, functional annotation

## Abstract

**Simple Summary:**

Detailed studies of bird microsatellite distribution patterns are scarce compared to other eukaryotes. Hence, we performed a comprehensive comparative analysis of microsatellite distribution patterns for 53 birds from 16 orders. We also explored the function of genes contained in microsatellites. Our results revealed that the distribution patterns of microsatellites were subject to weak phylogenetic constraints. The comprehensive analysis of microsatellites indicated that the abundance and diversity of perfect microsatellites were affected by their lengths. Finally, we found that perfect microsatellites were enriched at the ends of genes, and these genes were associated with signal transduction and cellular process.

**Abstract:**

Microsatellites (SSRs) are widely distributed in the genomes of organisms and are an important genetic basis for genome evolution and phenotypic adaptation. Although the distribution patterns of microsatellites have been investigated in many phylogenetic lineages, they remain unclear within the morphologically and physiologically diverse avian clades. Here, based on high-quality chromosome-level genomes, we examined the microsatellite distribution patterns for 53 birds from 16 orders. The results demonstrated that each type of SSR had the same ratio between taxa. For example, the frequency of imperfect SSRs (I-SSRs) was 69.90–84.61%, while perfect SSRs (P-SSRs) were 14.86–28.13% and compound SSRs (C-SSRs) were 0.39–2.24%. Mononucleotide SSRs were dominant for perfect SSRs (32.66–76.48%) in most bird species (98.11%), and A(n) was the most abundant repeat motifs of P-SSRs in all birds (5.42–68.22%). Our study further confirmed that the abundance and diversity of microsatellites were less effected by evolutionary history but its length. The number of P-SSRs decreased with increasing repeat times, and longer P-SSRs motifs had a higher variability coefficient of the repeat copy number and lower diversity, indicating that longer motifs tended to have more stable preferences in avian genomes. We also found that P-SSRs were mainly distributed at the gene ends, and the functional annotation for these genes demonstrated that they were related to signal transduction and cellular process. In conclusion, our research provided avian SSR distribution patterns, which will help to explore the genetic basis for phenotypic diversity in birds.

## 1. Introduction

Microsatellites (simple sequence repeats, SSRs) are short tandem repeats of 1–6 nucleotide DNA motifs [1], which are functionally important for chromatin organization, recombination, gene transcription, translation, and DNA replication [2]. Microsatellites are widely distributed in both coding and noncoding regions of eukaryotic and prokaryotic genomes [3] but display a non-random distribution in the different genomic regions [4,5]. Due to their high levels of polymorphism [2], SSRs have become an important source of genetic divergence that can lead to a variety of phenotypic differences and provide a basis for adaptation to different environments [6,7]. For example, the accumulation of repetitive DNA sequences, including microsatellites, has been linked to the enlargement of the W chromosome [8,9,10,11] and Z chromosome [12], which may have led to the differentiation of sex chromosomes [13] and evolution of a karyotype [14,15]. Therefore, a comprehensive analysis of microsatellites across the evolutionary landscape can help identify functionally relevant SSRs [16] and determine their roles in the adaptive evolution of organisms [4].

Comparative analyses of microsatellite distribution patterns have mostly been limited by the quality of the SSRs datasets [17]. The SSRs identification and inferences are mainly affected by the accuracy of genome assembly and the completeness of sequence information [16]. Fortunately, advances in chromosome level assembly technology have improved the accuracy of identification for microsatellites within the whole genome. Furthermore, microsatellite studies based on high-quality chromosome-level genomes may contribute to understanding the connection between chromosome length and SSR number, abundance, density and GC content [18]. In fact, microsatellite recognition based on chromosome-level genomes have been applied to a variety of vertebrates, including Rhesus monkeys [19] and bovid species [18].

Birds (Aves) represent a monophyletic vertebrate clade that contains >10,000 species, and flight has allowed many birds to span the world and evolve considerable morphological and physiological diversity [20]. Despite the vast phenotypic diversity of birds, birds have an extremely compact genome with a small amount of repetitive DNA (4–10%) [21]. Compared to other vertebrates, birds have the lowest average SSR density with very little variance [16]. These genome characteristics are thought to be related to the selective pressure from flight adaption that required a high rate of oxidative metabolism [22,23]. However, comparative analyses of genome-wide microsatellite distribution within birds are scarce [24,25]. The distribution patterns of microsatellites between orders or classification groups remain unclear and require further exploration. Therefore, studying the microsatellite distribution patterns of birds can provide important information to understand avian genetic structure and evolutionary history. However, published chromosome-level bird genomes are relatively rare, and large-scale comparative analyses of microsatellite distribution patterns among avian taxa have been poorly investigated.

With the development of DNA sequencing technology, high-quality chromosome-level genomes have been assembled for a wide range of birds, which provides an opportunity to evaluate and compare the microsatellite distribution patterns in birds. Here, we conducted a comparative analysis of SSRs for 53 avian taxa from 16 orders. We examined the distribution patterns of SSRs among different avian branches and analyzed the variation characteristics of perfect SSRs in different genomic regions (coding sequences (CDSs), exons, introns, and intergenic regions). Further, the functional annotation of genes whose coding regions contained SSRs were explored based on Gene Ontology (GO) and Kyoto Encyclopedia of Genes and Genomes (KEGG) pathway analyses. This study will increase our understanding about the biological significance of SSRs for birds and the underlying genetic basis of phenotypic variation.

## 2. Materials and Methods

### 2.1. Sources of Genomic Dataset

The high-quality chromosome-level genome sequences of 53 birds were obtained from NCBI (https://www.ncbi.nlm.nih.gov/ accessed on 30 May 2022). These 53 avian species represented 16 of the extant 41 orders of birds, and the detailed information is listed in Appendix A. We also downloaded the phylogenetic tree of 53 birds from the Timetree platform [26], which could provide evolutionary background information and help to compare the SSRs’ distribution patterns among different avian taxa.

### 2.2. SSRs Identification and Characterization

We used Krait v1.0.3 [27] to identify P-SSRs, C-SSRs, and I-SSRs, and localized their relative positions in exons, introns, and CDSs from the annotation files. Meanwhile, we used Python script to identify P-SSRs scattered in intergenic regions [28]. According to previous detection standards [19,28], the minimum repeat unit of mononucleotide, dinucleotide, trinucleotide, tetranucleotide, pentanucleotide, and hexanucleotide P-SSRs were defined as 12, 7, 5, 4, 4, and 4, respectively. The repetitive sequences with circular permutations and/or reverse complementarity were grouped together as the same type of microsatellites.

To evaluate the abundance of genomic SSRs, we used Python script to calculate the loci/Mb for 53 species. Subsequently, we analyzed the repeat motif preferences for six types of P-SSRs. Furthermore, we conducted the phylogenetically informed Principal Component Analysis (PCA) through the phylo.pca function in phytools to compare the repeat motif preferences among avian lineages [29]. The first two principal components (PC1 and PC2) were the main factors affecting the variation of P-SSRs motif types. To further identify the differentiation level for P-SSR motifs among avian linages [25], we demonstrated the heat maps of loci/Mb frequencies and average bp/Mb for each motif in di- and trinucleotide P-SSRs, and the 25 most common motifs in tetra-, penta-, and hexanucleotide P-SSRs.

### 2.3. Variation Analysis of P-SSRs

Studies on repeat copy number (RCN) of P-SSRs have been thought to help understand the processes of mutation and selection [30]. Therefore, we selected five widely studied model bird species, the red junglefowl (*Gallus gallus*), mallard (*Anas platyrhynchos*), budgerigar (*Melopsittacus undulatus*), zebra finch (*Taeniopygia guttata*), and golden eagle (*Aquila chrysaetos*), to examine the relationship between P-SSRs numbers and repeat times. Since GC content is associated with the stability of genomic structure [18], we used the Python script to analyze the GC content and the coefficient of variability (CV) of RCN for six types of P-SSRs and compared the difference between the six orders that contained greater than two species [5].

### 2.4. Functional Analysis

To describe the distribution patterns of P-SSRs in genes, we counted the number of SSRs in the top 30 exons/introns at both ends of each gene, based on the genomes of the five commonly studied birds (i.e., *G. gallus*, *A. platyrhynchos*, *M. undulatus*, *T. guttata*, and *A. chrysaetos*) (Figure 9). We then undertook functional annotation based on Gene Ontology (GO) and Kyoto Encyclopedia of Genes and Genomes (KEGG) for genes contained P-SSRs in coding regions to further understand the biological role of microsatellites (Figure 10).

## 3. Result

### 3.1. Characteristics of Avian SSRs

I-SSRs were the most frequent SSR (69.90–84.61%) of the three types of SSRs in birds, followed by P-SSRs (14.86–28.13%) and C-SSRs (0.39–2.24%) (Figure 1a and Appendix A). Among the avian taxa we studied, Anseriformes contained the highest proportion of P-SSRs (25.63–28.13%) in their genomes, while *Rhegmatorhina hoffmannsi* had the lowest proportion of P-SSRs (14.86%) (Appendix A). For six types of P-SSRs, mononucleotide P-SSRs were dominant (32.66–76.48%) in most bird species (98.11%), followed by tetranucleotide P-SSRs (6.21–19.33%), and the proportion of hexanucleotide P-SSRs was the smallest (0.56–3.45%) (Figure 1b). However, *Colaptes auratus* did not adhere to these patterns, whose pentanucleotide P-SSRs were dominant (36.77%). Abundance distribution patterns for mono-, di-, tri-, and tetranucleotide P-SSRs were similar to total P-SSRs among the 53 bird species, but penta- and hexanucleotide P-SSRs were not (Figure 2). The proportion of four types of short P-SSRs (mono-, di-, tri-, and tetranucleotide) among Anseriformes species was relatively consistent but varied in other species. There was a relatively weak significant phylogenetic constraint on the proportion of the six types of perfect microsatellites in bird genomes.

We identified the most abundant repeat motifs of P-SSRs (Figure 3). (A)n was the most common repeat motif of P-SSRs in all birds (25.42–68.22%), and the other type of P-SSRs were highly varied among different birds. It is worth noting that poly(A)-rich repeats were relatively common in every type of P-SSRs. Overall, the repeat motif preferences for P-SSRs varied among avian species.

To identify the P-SSR repeat motif preferences across the evolutionary landscape, we conducted a phylogenetic PCA for di-, tri-, tetra-, penta-, and hexanucleotide P-SSR motifs (Figure 4 and Figure 5). The results of the phylogenetic PCA showed a weak phylogenetic signal that the variation of P-SSR motif types could not be explained by the phylogenetic relationship among the taxa we studied. Based on the corresponding heat maps, the distribution of loci/Mb frequencies and average bp/Mb for five motifs in different species were relatively consistent. We found that each PC1 was driven largely by variation in the frequency of (AT)n, (AAT)n, (AAAC)n, (AAAAC)n, and (AAAAAG)n, consistent with the relatively high abundance and length of these motifs. These results indicated that poly (A)-rich motifs were ubiquitous in avian genomes. It is worth noting that these high frequency motifs were mainly found in Galliformes and Anseriformes species. We also found that the diversity of motifs decreased as the motif length increased, where longer P-SSR motifs tended to have more stable preferences in avian genomes.

### 3.2. The Variability of Repeat Copy Number

The statistical results showed the number of each type of P-SSRs significantly decreased as the repeat times increased, except the trinucleotide P-SSRs had a double peak in *A. platyrhynchos* (Figure 6). The second peak of trinucleotide P-SSR number appeared at 10 repeat times and then gradually decreased. Furthermore, we analyzed the CV of RCN for P-SSRs in different genomic regions (Figure 7) and found that the variation trend from mononucleotide to hexanucleotide P-SSRs varied between avian lineages and genomic regions. The CV of RCN for P-SSRs in all species showed an upward trend from mononucleotide to hexanucleotide throughout the whole genome, and similar patterns had been observed in exons and intergenic regions. However, the trend of the CV of RCN for P-SSRs in CDSs and intron regions was unimodal (Figure 7a) with the peak occurring in penta- and tetranucleotide P-SSRs, respectively. Longer motifs of P-SSRs were more likely to have a higher CV of RCN. Similar patterns occurred in the genomes of Anseriformes (Figure 7b) and Falconiformes (Figure 7d). However, few similarities could be found among Apodiformes (Figure 7c), Galliformes (Figure 7e), Passeriformes (Figure 7f), and Psittaciformes (Figure 7g), which indicated that the CV of RCN for P-SSRs varied between bird orders.

### 3.3. GC Content, and Functional Analysis

We calculated the GC content of P-SSRs in different genomic regions (Figure 8). The results demonstrated similar patterns in GC content of P-SSRs throughout the genome and in different regions of the genome, where penta- and hexanucleotide P-SSRs contained the highest GC content while the GC content of mono- and dinucleotide P-SSRs was low. We observed similar trends across bird taxa. These results suggested that the GC content of longer motifs is more likely to be higher than shorter motifs in both whole genome and different genomic regions. We also found that the GC content of P-SSRs was the highest in the CDS region (Figure 8b), indicating that P-SSRs were more conserved in the coding region. Subsequently, we counted P-SSR numbers in exons and introns for each gene and found that exons/introns near the ends of genes contained relatively greater SSRs, indicating that P-SSRs were abundant at both ends of genes (Figure 9). Further, we analyzed the function of genes that contained P-SSRs in the coding region. The results of KEGG pathway analysis showed (Figure 10a) that these genes were involved in 44 channels and mainly distributed in the pathway of environmental information processing, cellular processes, organismal systems, and human disease. Specifically, a large number of genes containing microsatellites were involved in the signal transduction of environmental information processing. The results of GO analysis demonstrated that these genes were involved in 31 pathways and mainly found in biological processes, including cellular processes of biological processes.

**Figure 8 animals-13-00655-f008:**
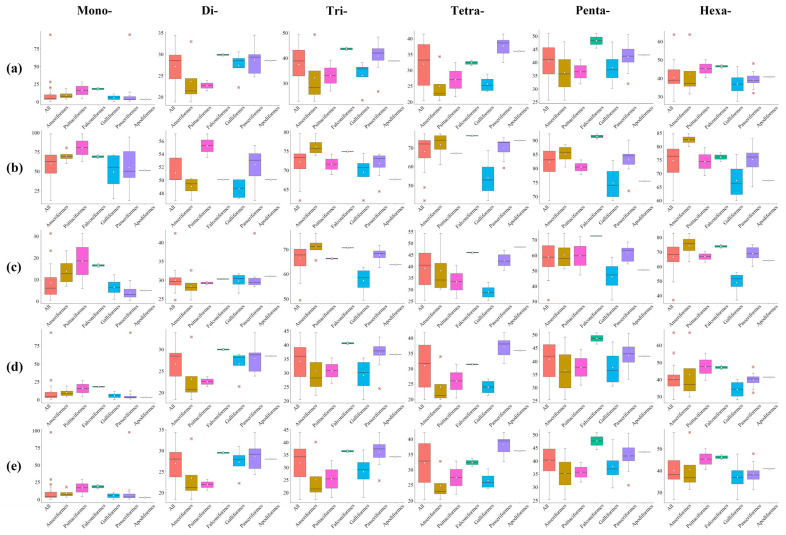
The GC content of P-SSRs in different genomic regions. (**a**) Genome; (**b**) CDS region; (**c**) Exon region; (**d**) Intron region; (**e**) Intergenic region. The small red squares are outliers.

**Figure 9 animals-13-00655-f009:**
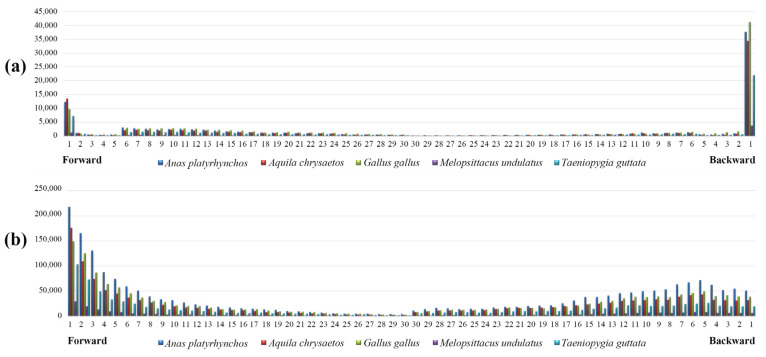
P-SSR distribution patterns in genes. (**a**) exons; (**b**) introns.

**Figure 10 animals-13-00655-f010:**
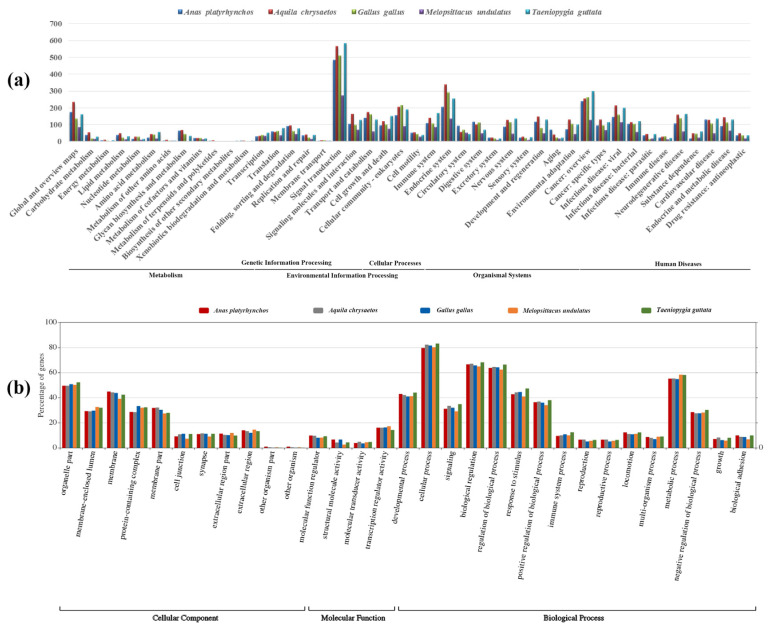
The result of functional annotation for genes contained P-SSRs in coding regions. (**a**) KEGG database (**b**) GO database.

## 4. Discussion

### 4.1. Abundance of SSRs in Avian Genomes

In present study, we identified and analyzed microsatellite distribution patterns based on chromosome-level genomes of 53 birds. The results showed that I-SSRs were the most abundant SSR in avian genomes, followed by P-SSRs and C-SSRs. The higher abundance of I-SSRs is common in the genomes of different lineages, including beetles [4] and Euarchontoglires [31]. Imperfect SSRs were caused by substitutions, insertions, and deletions of perfect SSRs [32], and the high density of imperfect SSRs in the human genome is the result of disrupting mutation accumulation [33]. Indeed, imperfect SSRs were thought to play an important role in regulating transcription and replication or modifying the structure of proteins [34]. For example, I-SSRs in the coding region have been demonstrated to prevent coding region frameshifts caused by microsatellite instability [35] and affect mismatch repair activity [36]. The I-SSRs concentrated at certain locus in potyvirus genomes may provide genetic variation through recombination and drive host adaptation to variable environments [37]. Therefore, the prevalence of I-SSRs in avian genomes may reflect that imperfect SSRs were an important genetic variation resource for the vast phenotypic diversity of birds.

Our study found that mononucleotide P-SSRs were the most abundant SSR in avian species we studied, which was consistent with previous studies in eukaryotic genomes, especially in birds and mammals [3,16,18,38]. (A)n was the most common repeat motif of P-SSRs in all birds (25.42–68.22%) (Figure 3), which indicated that birds indeed harbor highly abundant poly(A)-rich repeats with a low GC content [16,24]. The high frequency of poly (A) may be the result of insertion of processed genes into the genome from mRNA with a long poly(A) tail, which was necessary for the universal retrotransposon in eukaryotic genomes [39]. The low GC content of SSRs is the character of eukaryotic genomes, especially in birds [3], that can be explained on the basis of genomic GC content [16] and the relative difficulty of strand separation for GC compared to AT [40]. However, it is worth noting that all types of P-SSRs had the highest GC content in the coding region (Figure 8b), which was similar to bovid genomes [30]. The relatively high GC content of P-SSRs in the coding regions may affect the genome structure [18], methylation pattern [41], and gene expression [42]. Therefore, considering the conservatism of the GC base pair, it is reasonable to speculate that a GC-rich sequence may promote the normal gene expression by reducing the diversity of SSRs.

### 4.2. Distribution of SSRs across Avian Lineages

Different distribution patterns of P-SSRs among taxonomic vertebrate clades have been found in previous studies [16,25,43], while differences among taxa within avian lineages remains unclear [24]. The results of the statistical analysis and the phylogenetic PCA showed a weak phylogenetic correlation between the variation of P-SSRs and the evolutionary history of birds. The distribution of P-SSRs in different avian taxa was varied except in Anseriformes species (Figure 1), which was consistent with a previous study [24]. FISH analysis based on microsatellite probes had demonstrated that the accumulation of microsatellites was species-specific and might occur independently in reptiles and birds [13,14]. We also found that the distribution of penta- and hexanucleotide P-SSR motifs were more irregular than shorter P-SSRs motifs. A possible explanation is that longer microsatellites may have experienced different selection pressures [43,44] and produced inconsistent patterns with shorter P-SSRs. For microsatellites with the same length, the repeat unit of shorter motifs was more than longer motifs. More repeat units result in more opportunities for replication slippage and might be a major influence in different selection forces between short and long motifs [43].

We found that P-SSRs were abundant at both ends of genes regardless of being in exons or introns (Figure 9). This may be the result of the biophysical constraints of protein structure, which limits the distribution of microsatellites in the middle of the protein [45]. Microsatellites located in genes were thought to provide a molecular basis for species to rapidly adapt through modifying genes and affecting the evolution of protein structure and function [2,6,46]. However, we caution attributing phenotype variation to microsatellites, because the number of microsatellites in each genes was limited [28]. On the whole, the distribution of P-SSRs in organism genomes was subject to multiple selective pressures, including length constraints and protein structure, and these effects together promote the adaptive evolution of organisms.

### 4.3. Length Variation of P-SSRs

Microsatellites are composed of tandemly repeated units of DNA [1], which means the length of microsatellites increase with increasing repeat times or RCN. The variation of RCN for microsatellites was an important source of genetic variation that can provide phenotypic difference and help organisms to adapt to the environment [2,6,7]. Generally, the abundance of microsatellites decreases as RCN increases in eukaryotes [1,5,31,47]. As expected, our analysis of microsatellite RCN in birds demonstrated a similar pattern; the number of perfect microsatellites significantly decreased as the repeat times increased. A persuasive model of microsatellite evolution suggests that the genome-wide distribution of microsatellite repeat length is the result of the balance between length mutation and point mutation [48,49]. The mutation rate of microsatellites generally increases with increasing repeat counts [43,50], whereas point mutations break long repeat arrays into smaller units. These complex interactions set an upper boundary for the growth of microsatellite length, and reaching the threshold would lead to large deletions or reparation [51]. Previous studies showed that longer SSRs may be more polymorphic [52] and unstable [36] than shorter SSRs. For example, replication errors at (G)16 repeat were 30-fold higher than (G)10 in human colorectal cells [36]. Furthermore, the RCN of P-SSRs has been shown to influence gene expression [53,54]. Therefore, it can be inferred that length constraints were the result of a protective mechanism, which could reduce the diversity and instability of SSRs and ensure the normal gene expression.

We found that the diversity of P-SSRs was not only associated with the total length, but also the length of motifs. The CV analysis of RCN for genomic P-SSRs showed an upward trend from mono- to hexanucleotide in most birds, which has not been reported in previous studies [4,31]. This trend might be attributed to the functional differences in different types of P-SSRs. For example, the mutation of trifolds, including tri- and hexanucleotide motifs, did not lead to frameshift mutations and caused a slight effect on gene expression [55]. While the mutation of non-trifold motifs would cause changes in gene expression and protein structure, it might also undergo stronger selective constraints and be more conserved in RCN variation [44]. Meanwhile, the results of the phylogenetic PCA demonstrated that the diversity of motifs decreased as the motif length increased, which means longer motifs tended to have specific motifs in avian genomes. The accumulation of specific motifs for different microsatellite types might be the result of selective constraints [43]. Because the preference of specific motifs may have specific effects on genome function [52], it affects the structure and function of proteins via higher frequencies of transcription [56]. In brief, the variation of total sequence and repeat motif lengths was the main driver of microsatellite diversity that may prevent the genome from being too redundant. Length constraints may help to retain more valuable genes in finite length sequences, reduce the energy cost of transcription and splicing, and improve the efficiency of gene expression. Therefore, organisms can quickly adapt to the changeable environment, especially in widely distributed bird species that span the world’s varied habitats.

## 5. Conclusions

Our results demonstrated that the genome-wide distribution of microsatellites was subject to weak phylogenetic constraints. The distribution patterns may be associated with the species-specific accumulation of microsatellites and indicate that the evolution of microsatellites are closely related to each species’ living environment. Meanwhile, our study confirms that length constraints were an important evolutionary force for microsatellites, as shown in the abundance and diversity of microsatellites decreasing with the increasing length. Finally, the identification and characterization of SSRs across avian evolutionary landscapes needs further exploration because it provides an opportunity to investigate the diversity of microsatellites in response to selection pressures from variable environments.

## Figures and Tables

**Figure 1 animals-13-00655-f001:**
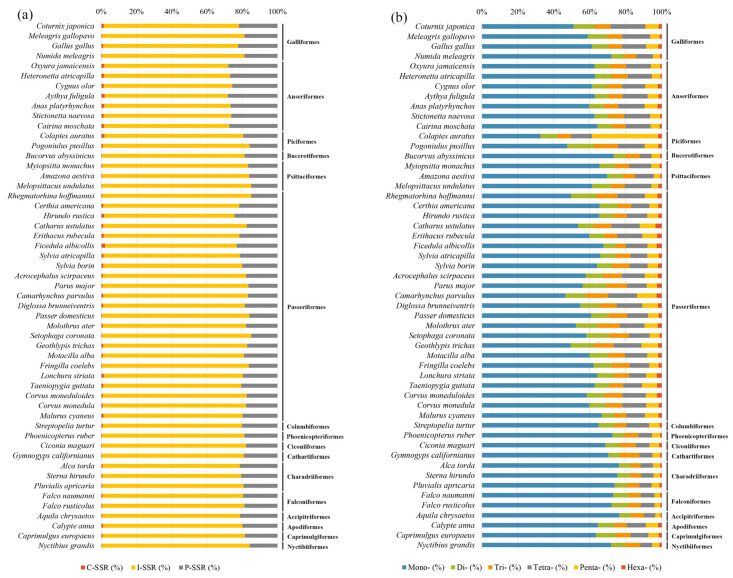
Proportion of each type of SSRs (**a**) and P-SSRs (**b**).

**Figure 2 animals-13-00655-f002:**
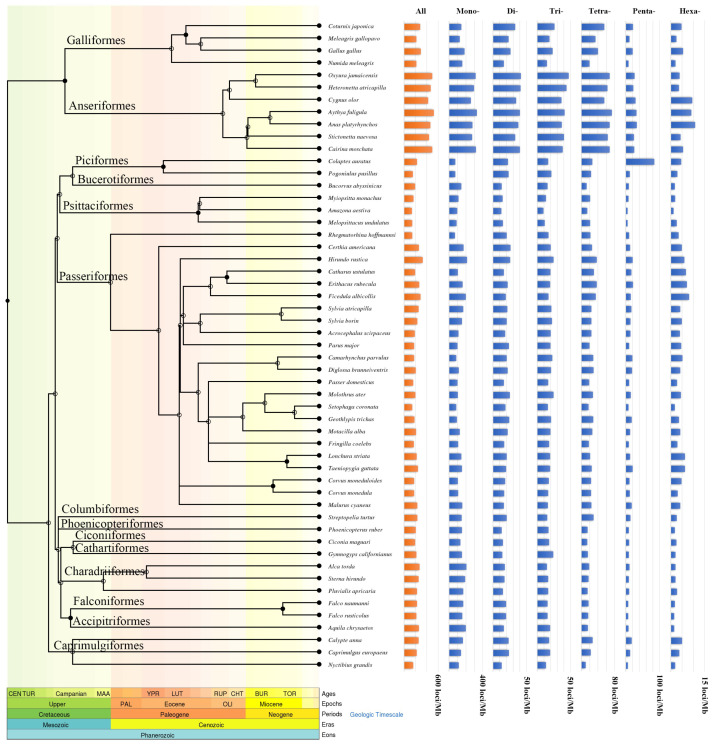
The abundance of P-SSRs.

**Figure 3 animals-13-00655-f003:**
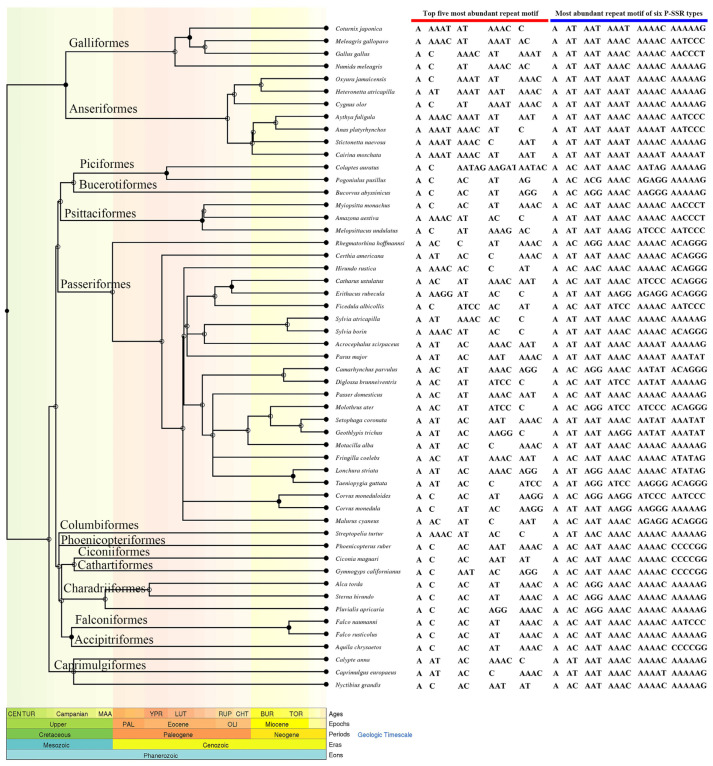
The repeat motifs preferences of P-SSRs.

**Figure 4 animals-13-00655-f004:**
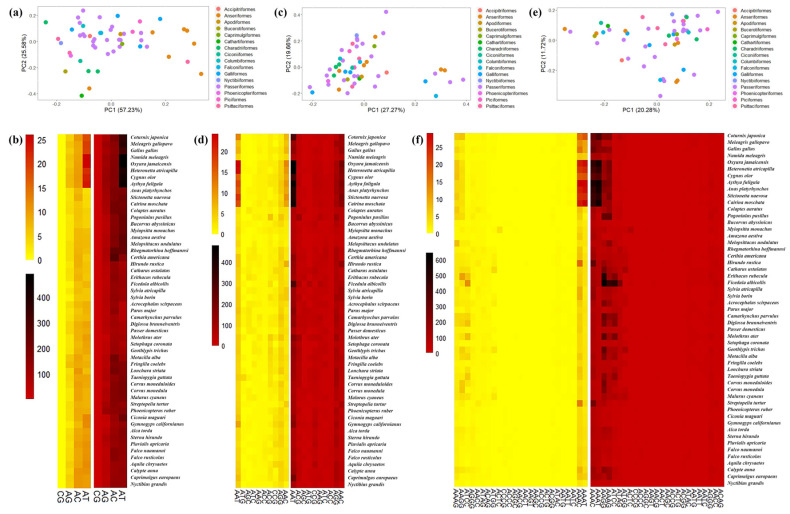
Phylogenetic PCA, loci/Mb heatmaps and bp/Mb heatmaps for di- (**a**,**b**), tri- (**c**,**d**), and tetranucleotide (**e**,**f**) P-SSR motifs. For all heatmaps, the yellow section represents the abundance (loci/Mb) of different motifs and the red section represents the density (bp/Mb) of each motif.

**Figure 5 animals-13-00655-f005:**
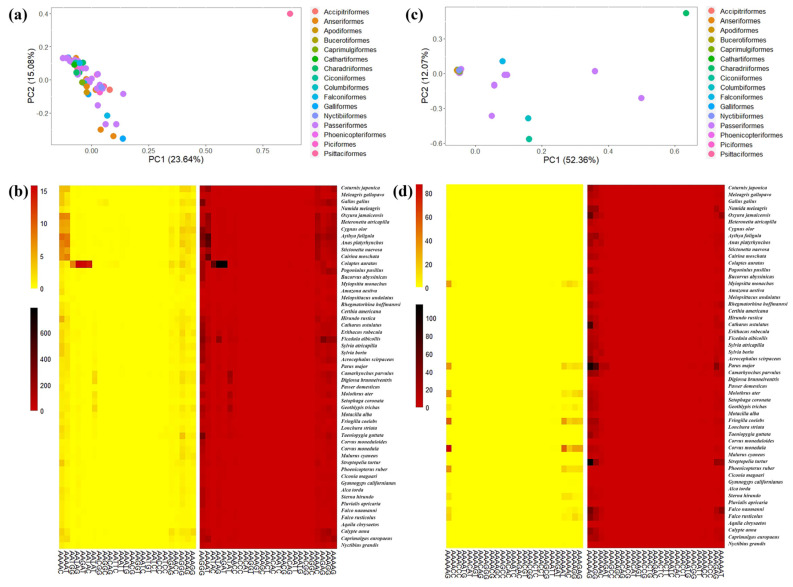
Phylogenetic PCA, loci/Mb heatmaps and bp/Mb heatmaps for penta- (**a**,**b**) and hexanucleotide (**c**,**d**) P-SSR motifs.

**Figure 6 animals-13-00655-f006:**
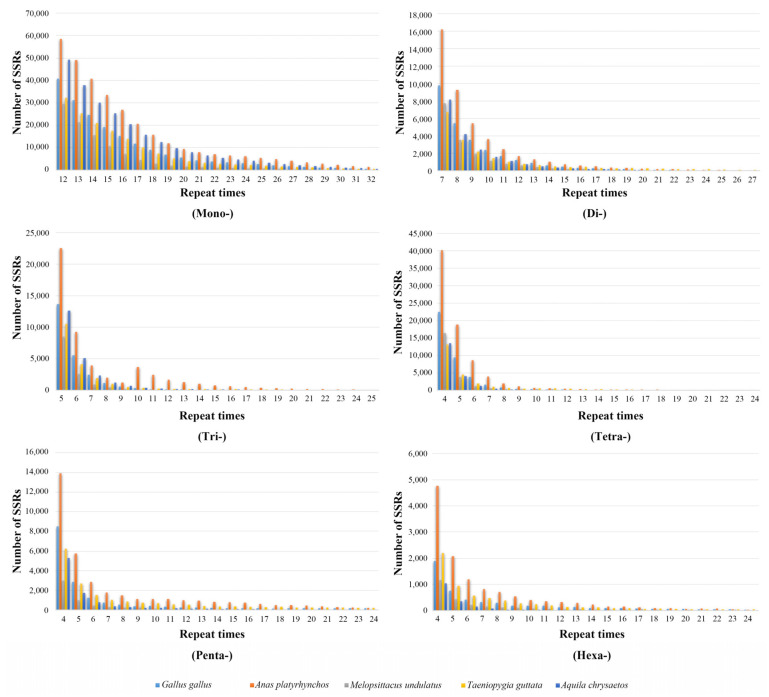
Relationship between repeat times and number of P-SSRs.

**Figure 7 animals-13-00655-f007:**
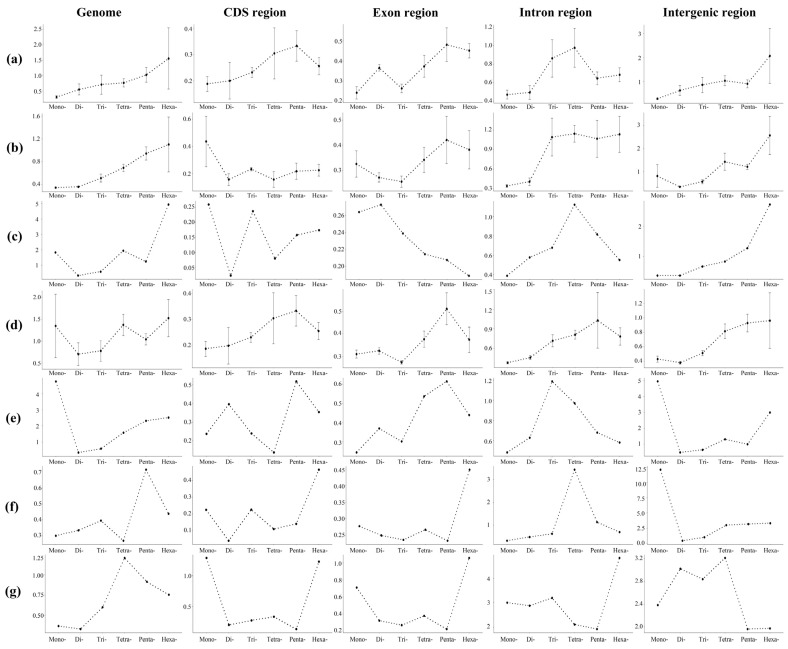
The coefficient of variability analysis of the repeat copy number of P-SSRs among six orders. (**a**) All; (**b**) Anseriformes; (**c**) Apodiformes; (**d**) Falconiformes; (**e**) Galliformes; (**f**) Passeriformes; (**g**) Psittaciformes.

## Data Availability

The data presented in this study are available in the article or Appendix A.

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
