# Peer review of "Comprehensive Comparative Analysis Sheds Light on the Patterns of Microsatellite Distribution across Birds Based on the Chromosome-Level Genomes"

_animals, 2023, doi:10.3390/ani13040655_

Round 1
Reviewer 1 Report
Dear Editor and authors,
I appreciate the invitation to review the manuscript and hope that my considerations can contribute to the improvement of the work and the dissemination of science. In the article, the authors carry out a genomic analysis in several species of birds, evaluating the pattern of distribution, size and variation of microsatellites.
The data is unprecedented. The text needs to be clarified in some points, which I will comment on later. My main questions are in the presentation of results.They could be clearer and the figures modified, as well as the captions more explanatory. Some figures are illegible, even after enlarging them, especially the figure 8. In each figure separate the orders by color. One suggestion is to place all the species of the same group with the font of the same color, for example.
This manuscript may be accepted after major modifications.
I put my considerations as comments below.
-
Correctly point out the two corresponding authors
-
Line 22: the sentence leaves doubt whether the three types of SSR had the same proportion in all taxa, that is, plus or minus 33% in each, which is wrong according to the results. I think the authors wanted to say that each type of SSR had the same ratio between taxa. Ex. Frequency of I-SSR was around X% at all taxa, while P-SSR around Y% and C-SSR Z%. I suggest rewriting the sentence
-
Line 78: explain the abbreviations GO and KEGG
-
Lines 94 to 96: were repetitions limited to 12, 7, 5, 4 and 4 or should the SSR have at least this number of repetitions to be considered an SSR, considering the size of each motif?
-
Lines 98 to 100 - Suggestion for change: “First, we used Python script to evaluate the abundance of genomic SSRs by calculating the loci/Mb for 53 species (Fig. 1a) and the abundance of each type of P-SSR (Fig. 1b and Fig. 2). After that we analyzed the most abundant repeat motif in general and the motifs preferences for each type of P-SSRs (Fig. 3).”
-
In addition, I suggest adding a table, with the values of each frequency: type of SSR, type of P-SSR, most abundant repeat motif and most abundant repeat motif of six P-SSRs.This will make it easier to compare the taxa.
-
Lines 129-130: Without the frequency values of each SSR it is difficult to compare just by looking at the graph because the bars are very similar. Also, is there a significant difference? Was any statistical test done? The number of samples is very different in each order, can this influence the result in relation to the distribution pattern within each group? Furthermore, in figure 2 the orders are not indicated, making it necessary to go back to figure 1. That is why I suggested that, in some way, the order to which that species belongs was indicated.
-
In lines 138-139, in my opinion, the proportions of only four categories were consistent within Anseriformes. Penta and hexa varied greatly among the species in the group.
-
Lines 200-201: Only by calculating the GC content is it possible to establish the function of the microsatellite in the genome? What did the authors mean by this sentence?
Reviewer 3 Report
This manuscript focuses on microsatellites in birds using a large genomic dataset. While I do not see many novel insights from this study, this study is still worth publication in some form. I appreciate the great efforts in analyzing 53 bird genomes, but I do not see why the genomes have to be chromosomo-level. Instead, genome quality matters more than whether they are chromosome-level, in the context of studying microsatellites. The authors did not provide information of whether the genome assemblies were derived from long-read or short-read sequencing, which will likely impact the identification of microsatellites and their abundance.
Some specific comments:
L28, what is CV?
L50-51, no, the chromosome-levle assemblies do not necessarily lead to complete identification. There will still be considerable gaps in assemblies if short-reads were used in contig assembly.
L59, Not fully accurate: not all birds are capable of flight.
L141 Figure 1, why the taxa were not ordered according to their phylogenetic positions like what is done in Figure 2
L145 Figure 2, please label the orders (or higher level classification) in the phylogenetic trees.
L170 Figure 3, I am not sure of what the PCA analyses can inform us.
L243, I do not understand how the authors come to the suggestion that I-SSRs were related to avian adaptive radiation. The previous contexts do not give such a clue.
L291, please be consistent in the use of terms. The term RCN could have been used in the results, and should be consistent throughout the manuscript.
L333-334, this is very speculative and is not supported by any of the analyses.
L339, this manuscript does not seem to deal with the polymorphism of microsatellites, so I am not sure how this sentence is relevant to this manuscript.
Round 2
Reviewer 1 Report
Dear Editor and Authors,
All suggestions or questions were appropriately answered. However, the figures remain illegible, even after they have been replaced. Maybe it's the way the image is exported from the program. I consider that the manuscript can be accepted, but I suggest that the authors try another way to improve the image quality.
Reviewer 3 Report
The authors have entirely ignored my previous major concern:
but I do not see why the genomes have to be chromosomo-level. Instead, genome quality matters more than whether they are chromosome-level, in the context of studying microsatellites. The authors did not provide information of whether the genome assemblies were derived from long-read or short-read sequencing, which will likely impact the identification of microsatellites and their abundance.
I would not recommend acceptance without the authors addressing my concerns.
Round 3
Reviewer 3 Report
I do not think the authors have addressed my concerns, and have not analyzed the impact of sequencing technology (short- or long-read) on the detection of microsatellites.